# Single-site catalyst promoters accelerate metal-catalyzed nitroarene hydrogenation

Liang Wang[1], Erjia Guan[2], Jian Zhang[1], Junhao Yang[3], Yihan Zhu[4], Yu Han [4], Ming Yang[5], Cheng Cen[5], Gang Fu[6], Bruce C. Gates[7] & Feng-Shou Xiao[1]

Atomically dispersed supported metal catalysts are drawing wide attention because of the opportunities they offer for new catalytic properties combined with efficient use of the metals. We extend this class of materials to catalysts that incorporate atomically dispersed metal atoms as promoters. The catalysts are used for the challenging nitroarene hydrogenation and found to have both high activity and selectivity. The promoters are single-site Sn on $TiO_2$ supports that incorporate metal nanoparticle catalysts. Represented as M/Sn-$TiO_2$ (M = Au, Ru, Pt, Ni), these catalysts decidedly outperform the unpromoted supported metals, even for hydrogenation of nitroarenes substituted with various reducible groups. The high activity and selectivity of these catalysts result from the creation of oxygen vacancies on the $TiO_2$ surface by single-site Sn, which leads to efficient, selective activation of the nitro group coupled with a reaction involving hydrogen atoms activated on metal nanoparticles.

[1] Key Laboratory of Applied Chemistry of Zhejiang Province, Department of Chemistry, Zhejiang University, Hangzhou 310028, China. [2] Department of Materials Science and Engineering, University of California, Davis, CA 95616, United States. [3] State Key Laboratory for Catalysis, Dalian Institute of Chemical Physics, Chinese Academy of Science, Dalian 116023, China. [4] Advanced Membranes and Porous Materials Center, Physical Sciences and Engineering Division, King Abdullah University of Science and Technology, Thuwal 23955-6900, Saudi Arabia. [5] Department of Physics and Astronomy, West Virginia University, Morgantown, WV 26506-6315, United States. [6] State Key Laboratory for Physical Chemistry of Solid Surfaces, and National Engineering Laboratory for Green Chemical Productions of Alcohols-Ethers-Esters, College of Chemistry and Chemical Engineering, Xiamen University, Xiamen 361005, China. [7] Department of Chemical Engineering, University of California, Davis, CA 95616, United States. These authors contributed equally: Liang Wang, Erjia Guan. Correspondence and requests for materials should be addressed to G.F. (email: gfu@xmu.edu.cn) or to B.C.G. (email: bcgates@ucdavis.edu) or to F.-S.X. (email: fsxiao@zju.edu.cn)

Metal catalysts dispersed on solid supports dominate the technology of production of chemicals, fuels, and polymers, and they are the keys to environmental protection by clean-up of effluent gases from motor vehicles and fossil fuel power plants[1–4]. Hydrogenations are among the important technological reaction catalyzed by supported metals[5–10]. An important example is the hydrogenation of substituted nitroarenes, a widely used route for producing functionalized anilines as intermediates for the production of agrochemicals and antidetonators and as building blocks for high-performance rubbers and polymers[11,12]. In nitroarene hydrogenation, it is difficult to control the selectivity when more than one reducible group is present in the reactant[10,13–24]. The hydrogenation usually occurs preferentially on the non-target groups, giving low yields of the desired functionalized anilines. Important progress toward improved selectivity has been achieved by using gold-, cobalt oxide-, and ferric oxide-containing catalysts[1,10,13–15], but these have the drawback of low activities, requiring high temperatures and long reaction times. Alternatively, the platinum-group metals offer the advantage of high catalytic activity, but their selectivities are generally low.

The performance of many prototypical supported metal catalysts (e.g., for industrial ammonia synthesis[25,26], Fischer–Tropsch synthesis[27,28], and the water-gas shift[29]) is enhanced markedly by promoters—additives that by themselves are not good catalysts. Understanding of the structures and roles of promoters markedly lags the understanding of catalysts. Important scientific foundations for understanding the roles of promoters have emerged from ultrahigh vacuum surface-science experiments with single metal crystals, but the results fall short of providing general guidance for the control of surface structures of supported catalysts.

We addressed this challenge in the context of selective hydrogenation of substituted nitroarenes and now report a general and efficient strategy to enhance the catalytic activity and/or selectivity of various metals (e.g., Au, Ru, Pt, and Ni) supported on TiO2—having achieved this goal by incorporating a single-site promoter—tin—on the TiO2 surface by a wet-chemical method giving what we refer to as metal/Sn–TiO2 catalysts. Our investigation illustrates that the Sn–O–Ti linkage facilitates the formation of oxygen vacancies on TiO2, which easily convert nitro groups into nitroso groups. Our promotion strategy gives a family of catalysts providing enhanced performance. For example, gold nanoparticles, which are generally selective but poorly active catalysts, as well as ruthenium, platinum, and nickel nanoparticles, which are active but poorly selective, are all transformed by the single-site tin promotion into catalysts that are both active and selective. Multi-technique characterization data show that the single-site tin species are associated with oxygen vacancies on titania formed in a hydrogen atmosphere that facilitate selective adsorption and activation of nitro groups, leading to the high catalytic activity and selectivity.

## Results

**Catalyst synthesis and characterization.** The tin sites were anchored to TiO2 (anatase) by grafting of dimethyl tin dichloride, followed by calcination to remove the methyl groups. Then gold nanoparticles were loaded onto the Sn-grafted anatase by a urea-assisted impregnation. The resultant catalyst, denoted Au/Sn–TiO2-123, had an Au loading of 0.7 wt% and a Ti/Sn atomic ratio of 123 (Supplementary Table 2).

The weak tin signals in energy dispersive X-ray spectra of Sn–TiO2-123 characterizing randomly selected nanoscale regions (Supplementary Fig. 1) indicate a high dispersion of tin on TiO2. The tin atoms remain highly dispersed after gold loading—the

**Table 1 EXAFS data characterizing Sn–TiO2-123 and Au/Sn–TiO2-123 under in situ treatment conditions**

| Sample | Shell | N | R (Å) | $10^3 \times \Delta\sigma^2$ (Å²) | $\Delta E_0$ (eV) |
|---|---|---|---|---|---|
| Sn–TiO2-123 | Sn–O₁ | 2.1 | 2.00 | 0.25 | −7.99 |
| | Sn–O₂ | 2.9 | 2.10 | 2.35 | 5.22 |
| | Sn–Ti₁ | 1.1 | 3.08 | 2.88 | 8.46 |
| | Sn–Ti₂ | 1.5 | 3.50 | 7.00 | 0.07 |
| Au/Sn–TiO2-123 | Sn–O₁ | 2.1 | 1.97 | 0.30 | −6.75 |
| | Sn–O₂ | 3.0 | 2.07 | 0.40 | 9.42 |
| | Sn–Ti₁ | 1.0 | 3.08 | 2.25 | 12.7 |
| | Sn–Ti₂ | 2.3 | 3.49 | 8.38 | 2.18 |
| H₂ treated Au/Sn–TiO2-123 | Sn–O₁ | 2.2 | 2.02 | 0.39 | −3.93 |
| | Sn–O₂ | 1.8 | 2.13 | 0.46 | 6.23 |
| | Sn–Ti₁ | 1.3 | 3.09 | 5.60 | 9.93 |
| | Sn–Ti₂ | 2.0 | 3.51 | 9.40 | 1.06 |
| Nitrobenzene-adsorbed on Au/Sn–TiO2-123 | Sn–O₁ | 2.3 | 1.98 | 0.10 | −7.99 |
| | Sn–O₂ | 3.0 | 2.09 | 1.32 | 7.90 |
| | Sn–Ti₁ | 1.0 | 3.07 | 5.05 | 6.94 |
| | Sn–Ti₂ | 2.4 | 3.49 | 13.0 | 6.40 |

EXAFS parameters characterizing Sn–TiO2-123 and Au/Sn–TiO2-123 (range of $k$ = 3.44–12.08 Å⁻¹, range of $R$ = 0.5–4.0 Å); $N$, coordination number; $R$, distance between absorber and backscatterer atoms; $\Delta\sigma^2$, disorder term; $\Delta E_0$, inner potential correction. Error bounds (accuracies) characterizing the structural parameters obtained by EXAFS spectroscopy are estimated to be $N$, ±15%; $R$, ± 0.02 Å; $\Delta\sigma^2$, ± 20%; $\Delta E_0$, ± 20%

lack of peaks characteristic of SnO2 or metallic gold in the XRD pattern of Au/Sn–TiO2-123 (Supplementary Fig. 2a) indicates that gold and tin species are both highly dispersed on the TiO2. The existence of Sn species on the surface of anatase was confirmed by X-ray photoelectron spectroscopy (Supplementary Fig. 2b). High-angle annular dark-field scanning transmission electron microscopy images of Au/Sn–TiO2-123 (Supplementary Fig. 3a and b) show gold nanoparticles with a mean diameter of 4.0 nm—clearly distinguished from the TiO2 support by the Z-contrast; tin species are not observable (Supplementary Fig. 3b and c), consistent with their being extremely small.

The average coordination environments of tin in Sn–TiO2-123 and in Au/Sn–TiO2-123 were determined by extended X-ray absorption fine structure (EXAFS) spectroscopy, with data recorded at the Sn–K edge (Table 1, Supplementary Fig. 4). In contrast to the spectra of the references, tin foil and bulk SnO2 crystals (Supplementary Table 3), Sn–Sn contributions are absent from the spectra of both Sn–TiO2-123 and Au/Sn–TiO2-123, consistent with the presence of tin in isolated single sites. Correspondingly, the X-ray absorption near edge structure data indicate the presence of tin as cationic species (Supplementary Fig. 5). Two Sn–O shells, with coordination numbers of approximately 2 and 3, were found and denoted as Sn–O₁ and Sn–O₂, respectively (Table 1 and Fig. 1). The Sn–O₁ and Sn–O₂ contributions are assigned to Sn atoms bonded to oxygen atoms via different Sn–O–Ti linkages. Notably, the Sn–O₁ bond length (2.0 Å) is less than that in bulk SnO2 crystals (2.05 Å, Supplementary Table 3), but the Sn–O₂ bond length is greater (2.1 Å).

Thus, to summarize, the EXAFS results provide evidence of single-site Sn in the Sn–TiO2-123 sample. Importantly, the single-site Sn is stable after loading of gold nanoparticles onto Sn–TiO2-123; the data show that single-site Sn in Au/Sn–TiO2-123 is in a coordination environment similar to that in Sn–TiO2-123 (Table 1).

**Catalyst performance.** Our investigation of the catalytic properties of Au/Sn–TiO2-123 began with the hydrogenation of

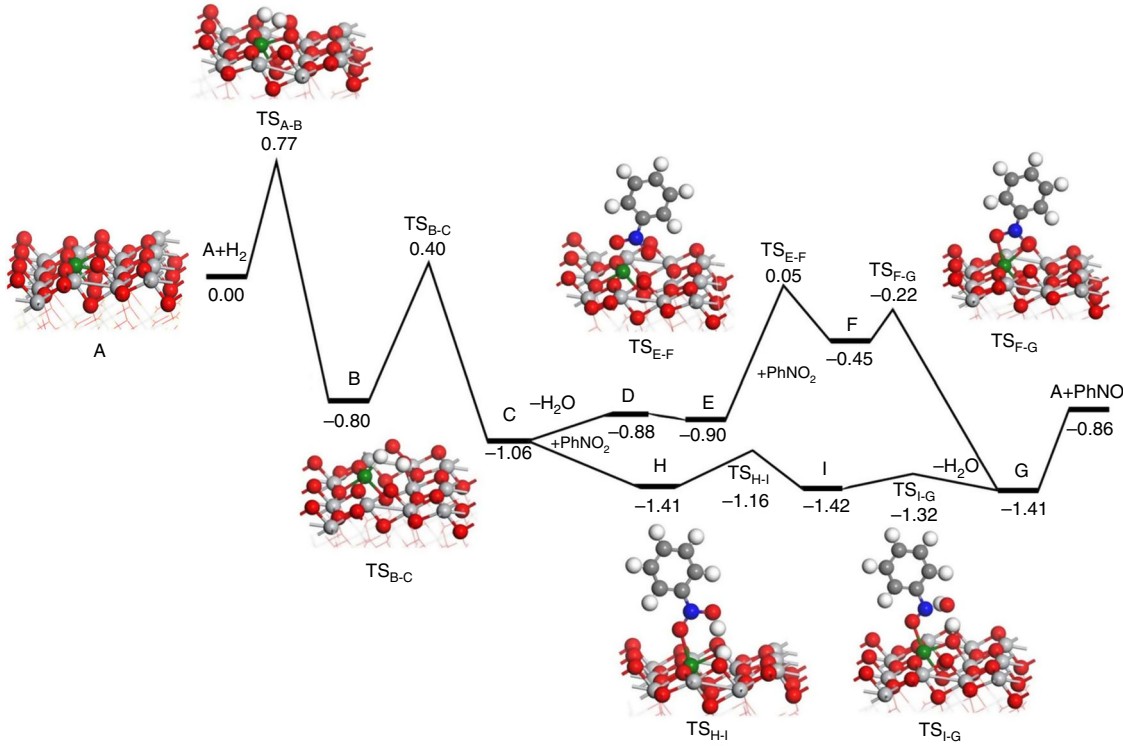

**Fig. 1** Energy profile of catalytic deoxygenation of nitrobenzene on $Sn_1/TiO_2(101)$ surface. Color index: Ti, gray; Sn, green; O, red; C, dark gray; H, white; N, blue

| Table 2 Catalytic data characterizing various Au catalysts in the hydrogenation of nitroarenes | | | | | |
|---|---|---|---|---|---|
| Catalyst | T (K) | Time (h) | Conversion (%) | Selectivity[a] (%) | N balance closure[b] (%) |
| Nitrobenzene reactant[c] | | | | | |
| $Sn–TiO_2$ | 373 | 1.5 | —[d] | — | >99.5 |
| $Au/Sn–TiO_2$-123 | 373 | 1.5 | >99.5 | >99.5 | >99.5 |
| $Au/TiO_2$ | 373 | 1.5 | 28.4 | >99.5 | >99.5 |
| $Au/TiO_2$ | 373 | 10.0 | 80.0 | >99.5 | >99.5 |
| $Au/Sn–TiO_2$-20 | 373 | 1.5 | 29.1 | 98.7 | 99.1 |
| $Au/SnO_2$ | 373 | 10.0 | 52.0 | 99.1 | 99.0 |
| 3-Nitrostyrene reactant[e] | | | | | |
| $Au/TiO_2$ | 353 | 4.0 | 18.9 | 93.9 | >99.5 |
| $Au/TiO_2$ | 353 | 20.0 | 79.0 | 92.5 | 98.9 |
| $Au/Sn–TiO_2$-123 | 353 | 4.0 | 99.0 | 99.3 | >99.5 |
| $Au/Sn–TiO_2$-20 | 353 | 20.0 | 24.2 | 91.2 | >99.5 |
| $Au/SnO_2$ | 353 | 20.0 | 18.0 | 75.1 | >99.5 |
| $Pt/TiO_2$ | 318 | 2.0 | 91.1 | 50.0 | >99.5 |
| $Pt/Sn–TiO_2$-123 | 318 | 2.0 | 98.5 | 97.4 | >99.5 |
| $Ru/TiO_2$ | 383 | 2.0 | 71.4 | 66.6 | >99.5 |
| $Ru/Sn–TiO_2$-123 | 383 | 2.0 | 99.0 | 98.4 | 98.5 |
| $Ni/TiO_2$ | 393 | 4.5 | 5.7 | 51.7 | 98.5 |
| $Ni/Sn–TiO_2$-123 | 393 | 4.5 | 52.0 | 90.1 | 99.2 |

[a] Selectivity to the functionalized aniline
[b] Calculated from the number of N-containing molecules in the reactor before and after the reaction
[c] Reaction conditions: 0.5 mmol of nitroarene, 40 mg of catalyst, 4 mL of toluene, 1.3 MPa of $H_2$
[d] Undetectable
[e] Reaction conditions: 0.5 mmol of nitroarene, 40 mg of catalyst, 4 mL of toluene, 0.2 MPa of $H_2$ for Pt catalysts, 1.3 MPa of $H_2$ for Ru catalysts and 2.5 MPa of $H_2$ for Ni catalysts

nitrobenzene as a prototype reaction (Table 2). The data show that the Sn–TiO2-123 sample without gold is inactive for the reaction, whereas Au/Sn–TiO2-123 gave complete nitrobenzene conversion in a short time of 1.5 h in a batch reactor at 373 K, with aniline as the sole product. In contrast, the Au/TiO2 sample without tin gave a nitrobenzene conversion of only 28.4% under the same conditions. Even after 10 h, the nitrobenzene conversion with Au/TiO2 was only 80.0%. Because these two catalysts had comparable gold loadings and nanoparticle sizes and the same TiO2 support (Supplementary Figs 3 and 6), we conclude that the

enhanced catalytic activity of Au/Sn–TiO$_2$-123 results from the presence of isolated Sn sites. Increasing the tin loading of Au/Sn–TiO$_2$ to give a higher Sn/Ti ratio of 1/20 led to a significant decrease in the conversion of nitrobenzene, which is attributed to the formation of ineffective bulk SnO$_2$ crystals, as shown by the XRD pattern (Supplementary Fig. 7).

When SiO$_2$ was used instead of TiO$_2$ as the support for gold nanoparticles, the resultant single-site Sn-modified supported gold catalyst (Au/Sn–SiO$_2$-129, Supplementary Figs 8 and 9) was found to have a catalytic activity similar to that of the tin-free catalyst (Au/SiO$_2$), demonstrating an essential role of TiO$_2$ with the single-site Sn for achieving a highly active gold catalyst.

The selective hydrogenation of the nitro group in the presence of various other reducible substituent groups on the aromatic ring, including carbonyl, chloride, amide, vinyl, or nitrile, was investigated in the range 353–373 K by characterizing the hydrogenation of 3-nitrostyrene, 4-nitrobenzaldehyde, 4-nitrochlorobenzene, 2-chloro-4-nitrophenol, 4-nitrobenzamide, and 3-nitrobenzonitrile (Table 2 and Supplementary Table 4), respectively. The Au/TiO$_2$ has been reported to be selective for the hydrogenation of 3-nitrostyrene[1], with a selectivity of 93.9% to 3-vinylaniline at a conversion of 18.9% (3-ethylaniline was a byproduct). In contrast, 3-ethylaniline was barely detectable in the reaction catalyzed by Au/Sn–TiO$_2$, giving a 3-vinylaniline selectivity of 99.3%. Significantly, the conversion of 3-nitrostyrene was much higher with Au/Sn–TiO$_2$-123 (99.0%) than with Au/TiO$_2$ (18.9%) under the same conditions, and the enhanced activity was further confirmed in our kinetics investigation (Supplementary Table 5). In the hydrogenation of other substituted nitroarenes, 2-chloro-4-nitrophenol, 4-nitrobenzaldehyde, 4-nitrochlorobenzene, 4-nitrobenzamide, and 3-nitrobenzonitrile, the Au/Sn–TiO$_2$-123 catalyst always exhibited much higher activity than Au/TiO$_2$. Even more significantly, the high selectivity to the corresponding substituted anilines was still maintained, or even further improved, with the promoted catalysts. These results demonstrate the excellent catalytic performance of Au/Sn–TiO$_2$-123 and a decisive role of the site-isolated tin.

The efficient hydrogenation of nitroarenes promoted by single-site tin is not limited to gold catalysts; we extended the observations to platinum, ruthenium, and nickel catalysts, which are known to be active but poorly selective for the hydrogenation of substituted nitroarenes. For example, in the hydrogenation of 3-nitrostyrene (Table 2), TiO$_2$-supported platinum nanoparticles (Pt/TiO$_2$) exhibited a 3-nitrostyrene conversion of 91.1% with a 3-vinylaniline selectivity of 50.0%, and the Sn–TiO$_2$-123-supported platinum nanoparticles (Pt/Sn–TiO$_2$-123) gave a significantly increased 3-vinylaniline selectivity of 97.4% with a 3-nitrostyrene conversion of 98.5%. The kinetics data also demonstrate the enhancement of both activity and selectivity in the reaction catalyzed by Pt/Sn–TiO$_2$-123 (Supplementary Table 5). In the hydrogenation of 2-chloro-4-nitrophenol (Supplementary Table 6), the Pt/Sn–TiO$_2$-123 catalyst gave complete conversion of 2-chloro-4-nitrophenol with a 2-chloro-4-aminophenol selectivity of 99.0%. Under the same conditions, the Pt/TiO$_2$ exhibited a much lower conversion (81.9%) and selectivity (56.0%) than the Pt/Sn–TiO$_2$-123. Similar results were observed in the selective hydrogenation of 3-nitrostyrene (Tables 2) and 2-chloro-4-nitrophenol (Supplementary Table 6) with nickel and ruthenium catalysts, whereby the Ru/Sn–TiO$_2$-123 and Ni/Sn–TiO$_2$-123 always exhibited much higher selectivity than the conventional Ru/TiO$_2$ and Ni/TiO$_2$ catalysts. These results confirm the high efficiency and generality of the role of promotion by single-site tin on TiO$_2$ in providing both highly active and highly selective TiO$_2$-supported metal catalysts for the hydrogenation of substituted nitroarenes.

**Spectroscopy of working catalyst.** To better understand the role of single-site Sn under reaction conditions, we did in-operando EXAFS spectroscopy characterizing H$_2$- and nitrobenzene-treatments of the catalysts working at 373 K. For example, we found that the Sn–O coordination number representing the longer Sn–O bond in Au/Sn–TiO$_2$-123 decreased significantly in the H$_2$ treatment, from 3.0 to 1.8 (Table 1 and Supplementary Fig. 10), implying that the weak Sn–O linkages are partially cleaved in this treatment. When nitrobenzene was then introduced to the H$_2$-treated Au/Sn–TiO$_2$-123 sample, this coordination number increased to 3.0, leading us to assign the contribution to nitro groups interacting with Sn sites (Fig. 1). Notably, the average length of this Sn–O bond formed after introducing nitrobenzene was only 2.09 Å, confirming the efficient interaction.

Further experiments were carried out with Au/Sn–TiO$_2$-123 by in situ Raman spectroscopy, which is widely used to investigate oxygen vacancies on TiO$_2$[30,31]. As shown in Supplementary Fig. 11a, Au/Sn–TiO$_2$-123 is characterized by the $E_g$ mode of anatase at 143 cm$^{-1}$, and this underwent a marked shift as a result of treatment in flowing H$_2$; after 30 min of treatment, the $E_g$ mode shifted to 152 cm$^{-1}$, indicating the removal of O sites to form oxygen vacancies, with changes in the Ti–O–Ti symmetry. For comparison, any shift of the $E_g$ mode was undetectable with the sample in flowing argon (Supplementary Fig. 12a). And, notably, for Au/TiO$_2$ in flowing H$_2$, the shift of the $E_g$ mode was slight (<4 cm$^{-1}$) (Supplementary Fig. 11b). Similar phenomena were also indicated by the in situ Raman spectra of Pt/Sn–TiO$_2$-123 (Supplementary Fig. 13). The removal of O sites from Au/Sn–TiO$_2$-123 and Pt/Sn–TiO$_2$-123 was confirmed by the mass signal of D$_2$O in the effluent gas observed for the sample treated in flowing D$_2$ (Supplementary Figs 14 and 15). When nitrobenzene was introduced to the H$_2$-treated Au/Sn–TiO$_2$-123 or Pt/Sn–TiO$_2$-123 (Supplementary Figs 11a and 13a), the $E_g$ mode shifted back to about 141–143 cm$^{-1}$, associated with interaction between oxygen atoms of nitro groups with the Sn–TiO$_2$ matrix—all in good agreement with the EXAFS results.

Furthermore, we emphasize that the single-site Sn is crucial for the formation of oxygen vacancies. When bulk SnO$_2$ crystals were present instead of single-site Sn (in Au/Sn–TiO$_2$-20), rare oxygen vacancies were formed, as confirmed by the in situ Raman spectra (Supplementary Fig. 16), D$_2$-reduction (Supplementary Fig. 17), and H$_2$-temperature programmed reduction (TPR, Supplementary Fig. 18) measurements, thus leading to a significant decrease in the conversion of nitrobenzene (Table 2). Further decreases in the Sn loading of Au/Sn–TiO$_2$-123 led to decreased catalytic activities (Supplementary Fig. 19), because of the decreased number of oxygen vacancies in the catalysts, as confirmed by the D$_2$-reduction measurements (Supplementary Fig. 20).

Moreover, we recorded IR spectra of nitrobenzene adsorbed on Au/Sn–TiO$_2$-123 to better understand the catalyst–support interactions. The as-synthesized Au/Sn–TiO$_2$-123 is characterized by weak bands at 1525 and 1346 cm$^{-1}$ (Supplementary Fig. 21a), assigned to an asymmetric stretching ($\nu_{asym}$) vibration of nitro groups adsorbed on Ti–OH groups and a symmetric stretching ($\nu_{sym}$) vibration of nitro groups adsorbed on surface Ti atoms, respectively[14,32,33]. In a temperature-programmed desorption at 373 K, these bands disappeared, indicating the weakness of the interactions between nitrobenzene and Au/Sn–TiO$_2$-123. Significantly, when nitrobenzene was adsorbed on H$_2$-treated Au/Sn–TiO$_2$-123, two bands again appeared, at 1525 and 1349 cm$^{-1}$ (Supplementary Fig. 21b), accompanied by a band at 1493 cm$^{-1}$, with a significant red shift from the conventional asymmetric stretching $\nu_{asym}$) vibration of nitro groups $\nu_{asym}$ (1525 cm$^{-1}$). The red shift indicates that the N=O bonds were significantly weakened to form nitroso group by interaction with the H$_2$-

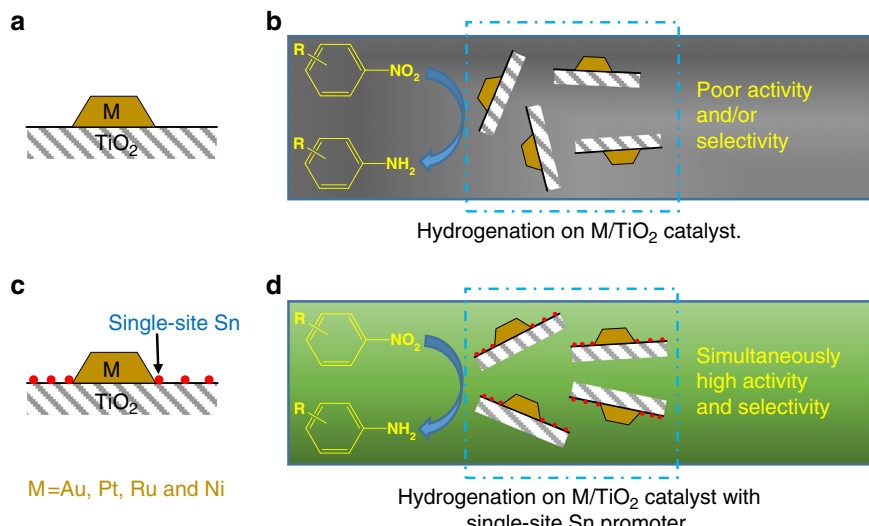

**Fig. 2** Model of catalysts with and without single-site Sn promotion. **a** Conventional M/TiO$_2$ catalysts; **b** M/TiO$_2$ catalysts in hydrogenation of substituted nitroarenes, for which poor activity or selectivity was obtained; **c** M/TiO$_2$ catalysts with single-site Sn promoters; **d** M/TiO$_2$ catalysts in hydrogenation of substituted nitroarenes, for which simultaneously high activity and selectivity were achieved by single-site Sn promotion

treated Au/Sn–TiO$_2$-123 catalyst. Even after desorption at 523 K, the Au/Sn–TiO$_2$-123 still exhibited a band at 1489 cm$^{-1}$, indicating that nitro or nitroso groups were adsorbed on the catalyst. Similar phenomena were also observed in experiments with the H$_2$-treated Pt/Sn–TiO$_2$-123 (Supplementary Fig. 22), in agreement with the results characterizing the Au/Sn–TiO$_2$-123 catalyst. Moreover, our data show that the adsorption of nitro groups on the Sn-modified catalyst was selective, hindering the adsorption of other groups (e.g., vinyl groups), as illustrated, for example, by IR spectra demonstrating that from a mixture of nitrobenzene and styrene, the former was strongly adsorbed, excluding the adsorption of the latter, thus leading to enhanced catalytic selectivity for the formation of vinyl aniline in the hydrogenation (Supplementary Fig. 23 and Supplementary Table 7).

Taking these results together with the in situ EXAFS, Raman, and IR spectra, we infer that the adsorption/interaction is associated with the presence of oxygen vacancies near the single Sn sites on Au/Sn–TiO$_2$-123. At the same time, the reactant hydrogen dissolved in or adsorbed on the metal nanoparticles readily facilitates the efficient and selective activation of nitro groups on the oxygen vacancies, which must be present near the metal nanoparticles—with the reaction taking place at the nanoparticle/support interface. These results confirm the importance of the support in Sn–TiO$_2$-123 and the new strategy for selective hydrogenation of substituted nitroarenes by employing single-site Sn promotion (Fig. 2).

**Catalyst recycle.** These catalysts are reusable. After each reaction experiment, the catalysts were easily separated from the reaction liquid, and, after simple washing, the catalyst was reused. In seven runs in the hydrogenation of 2-chloro-4-nitrophenol, the Au/Sn–TiO$_2$-123 exhibited 2-chloro-4-nitrophenol conversions and 2-chloro-4-amino phenol selectivities that were unchanged within error (Supplementary Fig. 24), indicating the good stability. Inductively coupled plasma (ICP)-optical emission spectrometry (OES) analysis of the reaction liquid after separation of the solid catalysts gave no evidence of detectable tin or gold species, indicating the absence of measurable leaching during the reaction. The high activity and selectivity combined with the good stability make these catalysts with single Sn promoter sites potentially valuable for application.

**Density functional theory calculations.** To gain deeper insight into the promotional effects of single-site Sn, we performed periodic density functional theory (DFT) calculations. It has been widely accepted that during the hydrogenation, nitrobenzene undergoes multistep reduction, successively producing nitrosobenzene, hydroxyaniline, and aniline. Our calculations demonstrate that when Sn is absent, the activity as well as the selectivity critically depend on the adsorption of nitrobenzene (Supplementary Figs 25 and 26, Supplementary Table 8). The adsorption energy of nitrobenzene on gold is rather weak (−0.04 ~ −0.27 eV), accounting for the low hydrogenation activity observed for the Au/TiO$_2$ catalysts. In contrast, nitrobenzene adsorbed flat on the Pt surfaces with its phenyl group (and potential substituent groups) strongly interacting with the Pt atoms (−1.05 eV ~ −2.10 eV). Strong adsorption could explain the high hydrogenation activity of Pt/TiO$_2$, and the parallel orientation of the aromatic ring could explain the low selectivity, because other substituent group could also be hydrogenated. Moreover, we found that the reduced intermediates and products would bond to the surfaces via their N atoms rather than their phenyl groups, suggesting that the hydrogenation would favor the formation of aniline once nitrosobenzene had formed. Thus, to enhance the selective hydrogenation, the promoter should facilitate the reduction of nitrobenzene to nitrosobenzene.

The calculations show that single-site Sn favors the generation of oxygen vacancies under the hydrogenation conditions (Supplementary Table 9), in agreement with the experimental results. Figure 1 shows the potential energy profile of a complete catalytic cycle on Sn$_1$/TiO$_2$(101) (Fig. 3 and Supplementary Fig. 27). H$_2$ activated heterolytically on the Sn$^{4+}$–O sites forms Sn(–H)–OH species (**B**), followed by H transfer to form an oxygen vacancy and adsorbed H$_2$O (**C**).

For the reduction of nitrobenzene, we considered two possible mechanisms, direct deoxygenation and water-assisted deoxygenation (Figs 1 and 3). In the former case, two adsorption states of nitrobenzene (**E** and **F**) on the oxygen vacancy were identified, with the interactions with the surface via the nitro groups rather than phenyl group, in contrast to the chemistry that occurs on noble metal surfaces[14,34,35]. Estimates of the Bader charges show that the nitrobenzene moiety in species **E** is nearly neutral (−0.02 a.u.), whereas in **F**, the nitrobenzene carries a charge of −0.56 a.u. —thus, **E** represents only physisorption whereas **F** represents a

**Fig. 3** Catalytic cycle. Proposed mechanism of catalytic deoxygenation of nitrobenzene on single Sn-substituted $TiO_2$ surfaces

charge-transfer state. The computations show that a conversion of **E** to **F** requires overcoming a barrier of 0.95 eV ($TS_{E-F}$), whereas that from **F** to **G**, the nitrosobenzene adsorption state, requires overcoming a barrier of only 0.23 eV ($TS_{F-G}$).

For water-assisted deoxygenation, the adsorbed $H_2O$ from the reaction was directly bound with $Ti^{4+}$, creating acidic groups. As shown in Fig. 1, nitrobenzene could be trapped by the adsorbed $H_2O$ though an OH···ONO hydrogen-bonding interaction, forming a physically adsorbed species (**H**). Next, **H** would undergo hydrogen transfer to give **I** ($TS_{H-I}$). Although **H** and **I** were found to be nearly the same in energy (−1.41 vs. −1.42 eV), they have different electronic configurations. Through charge analysis, we found that Sn still maintained the oxidation state of +2 in **H**, whereas Sn in **I** had been oxidized to $Sn^{4+}$. Thus, $TS_{H-I}$ can be viewed as having been formed by a proton coupling electron transfer in which electrons transfer from Sn to N, while a proton is transferred from $H_2O$ to the nitro group. In contrast to the formation of $TS_{E-F}$, the formation of $TS_{H-I}$ involves only a small barrier of 0.25 eV. This mechanism can be rationalized by considering that the adsorbed $H_2O$ stabilizes the negatively charged nitrobenzene during the electron transfer. In the next step, by overcoming a barrier of 0.20 eV, **I** would undergo another hydrogen transfer to produce adsorbed nitrosobenzene and reform the oxygen vacancy (**G**).

## Discussion

The combination of experimental and theoretical results leads to a clear picture of this new class of catalyst and its multiple functions. The theoretical results, consistent with experiment, indicate that the water-assisted deoxygenation mechanism is favored over the direct deoxygenation (Supplementary Figs 28 and 29). The selective hydrogenation of nitrobenzene thus benefits from the complementary roles of the catalyst components— the site-isolated Sn cations on the anatase support; the oxygen vacancies formed near these sites where the nitro groups are adsorbed; the nearby metal nanoparticles where reactant hydrogen is activated and made available to readily facilitate the generation of oxygen vacancies. These catalyst components act in concert and must be nearby each other, engaged in a mechanism

whereby $Sn_1/TiO_2$ facilitates the deoxygenation of nitrobenzene to nitrosobenzene, whereas the noble metal, in addition to facilitating $H_2$ dissociation, provides the sites for hydrogenation of nitrosobenzene intermediate to give aniline. We emphasize that the specific interaction between the $Sn_1/TiO_2$ support with its oxygen vacancies and reactant nitro groups is essential for the high activity and selectivity in the hydrogenation of nitroarenes on Sn–$TiO_2$-123 and the extended family of supported metal catalysts.

In summary, we present a strategy for improving the catalytic performance of a range of metal catalysts on conventional $TiO_2$ supports by employing single-site Sn sites on the $TiO_2$ as promoters. Gold, platinum, ruthenium, and nickel nanoparticles on the promoted support exhibit enhanced catalytic activity and selectivity for the hydrogenation of various substituted nitroarenes. Multi-technique characterization of the catalysts showed that the single-site Sn species are associated with oxygen vacancies formed in an $H_2$ atmosphere that facilitate selective adsorption and activation of nitro groups. The single-site Sn-promoted catalysts are stable and highly active and selective. The strategy of employing single-site promoters for selective supported metal catalysts may open the way to other catalysts with comparable combinations of components.

## Methods

**Catalyst synthesis**. Synthesis of Sn–$TiO_2$-123: In a typical experiment, 2.0 g of commercial anatase (20–50 nm) was initially heated at 100 °C for 10 h under vacuum in a 100 mL flask. After the addition of anhydrous toluene (60 mL, dried by $P_2O_5$) and dimethyltin dichloride (69 mg), the mixture was stirred at room temperature for 0.5 h. Then 5 mL of triethylamine was added, and the mixture was continuously stirred for another 2 h at room temperature. After filtration, washing with toluene and ethanol, drying, and treatment at 580 °C for 3 h in air, Sn–$TiO_2$-123 was finally obtained. The Ti/Sn atomic ratio of Sn–$TiO_2$-123 was 123. By increasing the amount of dimethyltin dichloride, Sn–$TiO_2$-20 was obtained. By using $SiO_2$ instead of $TiO_2$ as a support, Sn–$SiO_2$-129 with a Si/Sn ratio of 129 was obtained. By using 425 mg of dimethyltin dichloride in the starting solution, Sn–$TiO_2$-20 was obtained following the same procedures.

Synthesis of the supported Au and Pt catalysts: The supported metal catalysts were synthesized by the deposition-precipitation method. In a typical experiment for the synthesis of Au/Sn–$TiO_2$-123, 0.5 g of Sn–$TiO_2$-123 was added to 100 mL of $HAuCl_4$ (0.028 mmol of Au) and urea (2.9 mmol) solution. After stirring at 363 K for 4 h in a closed reactor kept in the dark, the liquid mixture was cooled in an ice bath at 273 K, followed by addition of $NaBH_4$ solution (400 mg of $NaBH_4$ in 20 mL of water). Finally, the solid sample was filtered and washed with a large amount of water, dried at 373 K for 12 h, and calcined at 473 K for 4 h. The Au/Sn–$TiO_2$-123 sample was then obtained.

**Catalytic reaction experiments**. The hydrogenation reactions were performed in a high-pressure autoclave with a magnetic stirrer (1000-1200 rpm). Typically, the reactant (substrate), catalyst, and solvent were mixed in the reactor and stirred for 15 min at room temperature. Then pure $H_2$ was introduced and kept at a desired pressure (measured at the reaction temperature), and the reaction system was heated to a given temperature (measured with a thermometer in the autoclave). After the reaction, the solid catalyst was separated from the fluid products, and the products were analyzed by gas chromatography (GC-14C, Shimadzu, with a flame ionization detector) with a flexible quartz capillary column coated with OV-17 or FFAP. The recyclability (reusability) of the catalyst was tested by separating it from the reactor liquid contents by centrifugation, washing with a large volume of methanol, and drying at 353 K for 6 h. In the recyclability test of Au/Sn–$TiO_2$-123, the gold concentration in the mother liquor was found to be less than 5 ppb by ICP analysis after separation of the used solid catalyst, indicating that essentially no gold leaching occurred during the reaction.

**Catalyst characterization**. X-ray photoelectron spectra were recorded with a Thermo ESCALAB 250 with Al Kα radiation at $h = 90°$ for the X-ray sources; the binding energies were calibrated by using the C1s peak at 284.9 eV. IR spectra were recorded with a Nicolet NEXUS 670 FT-IR spectrometer equipped with an MCT detector and ZeSe windows and a high-temperature reaction chamber. The nitrobenzene stream was introduced into the system in a flow of argon carrier gas. Raman spectra were recorded using a HR800 Raman spectrometer equipped with an Ar excitation source ($λ = 514.532$ nm). The nitrobenzene stream was introduced into the system with a flow of Ar carrier gas (30 sccm) with the reactant partial pressure in the range of 10–20 mBar. The catalyst metal content was determined by ICP analysis with a Perkin–Elmer Plasma 40 emission spectrometer and X-ray

fluorescence. High-resolution transmission electron microscopy imaging was carried out with a FEI-Titan ST electron microscope operated at 300 kV with a point resolution of 0.19 nm; the Au nanoparticle size distribution was determined by counting more than 100 nanoparticles.

Mass spectrometry: Mass spectra of the effluent gases introduced into a flow system or produced by reaction with the sample were measured with an online Balzers OmniStar mass spectrometer running in multi-ion monitoring mode. The sample was pressed into a thin wafer and loaded into a cell (In-situ Research Institute, South Bend, IN) through which helium continuously flowed at a rate of 100 mL/min. After the temperature had been increased to reach 393 K, $D_2$ was added to the helium stream with a flow rate of 20 mL/min. Changes in the intensities of major fragment of $D_2O$ ($m/z = 20$) were recorded. A bypass was used to determine whether $D_2O$ was formed in the sample. During the bypass, $D_2$ and helium flowed directly to the mass spectrometer without coming in contact with the sample. After switching the flow back to the line allowing gas to flow through the cell, a sharp increase in the $D_2O$ signal was observed, indicating that $D_2O$ was formed from the sample in the $D_2$-containing environment.

X-ray absorption spectroscopy: The X-ray absorption spectra were collected at beamline 4-1 at the Stanford Synchrotron Radiation Lightsource and at beamline (BL14W1) at the Shanghai Synchrotron Radiation Facility. A Si (220) or Si(111) double crystal monochromator was used and was detuned to 70% of maximum intensity to reduce the interference of higher harmonics present in the X-ray beam. The sample (approximately 0.6 g) was loaded into a vacuum sample holder and cooled to liquid nitrogen temperature during the measurement. For operando EXAFS spectroscopy, the sample was loaded into a flow-through cell and then was treated in flowing $H_2$ (10% $H_2$/He, flow rate at 20 sccm) or nitrobenzene (He flow through the liquid nitrobenzene at 353 K, flow rate at 20 sccm) at 373 K. The data were collected in transmission mode by use of ion chambers mounted on each end of the sample holder. A tin foil was placed down the beam of the second ion chamber and simultaneously measured as a reference.

The methods for data analysis of X-ray absorption spectroscopy and DFT calculations are available in the Supplementary Methods.

**Data availability**. All data are available within the article and its Supplementary Information file or from the authors upon request.

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

## Acknowledgements

This work was supported by the National Key Research and Development Program of China (2017YFC0211101) and National Natural Science Foundation of China (91645105, 21720102001 and 91634201). The work at University of California was supported by the U.S. Department of Energy (DOE), Basic Energy Sciences, grant number DE-FG02-04ER15513. We acknowledge beamtime at beamline 4-1 at Stanford Synchrotron Radiation Lightsource supported by the DOE Division of Materials Sciences, and beamtime at beamline BL14W1 at Shanghai Synchrotron Radiation Facility.

## Author contributions

L. Wang did the catalyst preparation, characterization, and performance testing experiments; E. Guan performed EXAFS experiments and data analysis as well as mass spectrometry experiments; J. Zhang participated in the catalyst characterization; Y. Zhu and Y. Han performed the TEM characterization; M. Yang and C. Cen provided helpful

discussions; J. Yang participated in the in situ EXAFS characterizations; G. Fu performed the DFT calculations; F.-S. Xiao, with the collaboration of B. C. Gates and G. Fu, conceptualized and planned this study, analyzed data, and wrote the paper with the collaboration of L. Wang and E. Guan.

## Additional information

**Competing interests:** The authors declare no competing interests.

