## [Peer Review File · Nature Communications]

Reviewers' comments:

Reviewer #1 (Remarks to the Author):

The authors have addressed most of the concerns of both referees by providing a large volume of new data. In particular, the DFT calculations provide an in-depth understanding of the reaction mechanism, and I like it. The revised manuscript is a high quality paper, and I am happy to recommend its publication in Nat Commun.

Reviewer #2 (Remarks to the Author):

The present authors showed Sn atoms on TiO₂ promotes the hydrogenation of nitro groups and discuss the mechanism of such promotion. However, they missed another important issue. That is the selectivity to vinyl aniline (VA) in the nitro styrene hydrogenation. M/Sn-TiO₂ (M = Au, Ru, Pt, Ni) showed very high VA selectivity (Table 2). This cannot explained simply the promotional effect of Sn on the hydrogenation of nitro groups. To get high VA selectivity, the hydrogenation of C=C should be retarded; however, this was not discussed. The authors should do this.

Point-by-Point Responses to the Comments

Reviewer #1

Comments: *The authors have addressed most of the concerns of both referees by providing a large volume of new data. In particular, the DFT calculations provide an in-depth understanding of the reaction mechanism, and I like it. The revised manuscript is a high quality paper, and I am happy to recommend its publication in Nat Commun.*

Responses: Thanks for the positive comments.

Reviewer #2

Comments: *The present authors showed Sn atoms on TiO₂ promotes the hydrogenation of nitro groups and discuss the mechanism of such promotion. However, they missed another important issue. That is the selectivity to vinyl aniline (VA) in the nitro styrene hydrogenation. M/Sn-TiO₂ (M = Au, Ru, Pt, Ni) showed very high VA selectivity (Table 2). This cannot explained simply the promotional effect of Sn on the hydrogenation of nitro groups. To get high VA selectivity, the hydrogenation of C=C should be retarded; however, this was not discussed. The authors should do this.*

Response: Thanks for the valuable comments. Yes, we agree that addition of information to address the point about hydrogenation of the C=C bond is appropriate as part of the assessment of the high selectivity in nitrostyrene hydrogenation. We had already been working on this and have data that we did not include in the original manuscript. Specifically, we performed competitive hydrogenation and adsorption experiments to characterize -NO₂ and -C=C groups on the TiO₂-supported Pt nanoparticle catalysts with and without Sn promotion.

We have added the new results to the revised manuscript, and the discussion has been expanded to address the reviewer's point.

Reviewer-Only-Table 1 presents the turnover frequency values (TOFs) calculated from the initial reaction rates as the number of molecules transformed per hour per Pt atom at various ratios of reactant (substrate) to catalyst (S/C) in the competitive hydrogenation of -NO₂ and -C=C groups by employing nitrobenzene and styrene as model reactant molecules for reaction on the TiO₂- supported Pt nanoparticle catalysts with and without Sn promotion. Notably, both Pt/TiO₂ and Pt/Sn-TiO₂-123 catalysts are active for the hydrogenation of nitrobenzene and styrene. However, when a mixture of nitrobenzene and styrene was used, the hydrogenation of styrene was strongly inhibited (TOF of 1262 with styrene reactant VS 134 with mixed reactants) in the presence of nitrobenzene on the Pt/Sn-TiO₂-123 catalyst. In contrast, the hydrogenation of styrene was only slightly influenced (TOF of 1608 with styrene reactant vs. 905 with mixed reactants) in the presence of nitrobenzene on the Pt/TiO₂ catalyst. These data confirm that the hydrogenation of -C=C group was strongly hindered by the presence of -NO₂ groups, and this result is in good agreement with the observed high selectivity for vinyl aniline in hydrogenation of nitrostyrene.

Furthermore, we studied the competitive adsorption of a mixture of nitrobenzene and styrene on the Pt/TiO₂ and Pt/Sn-TiO₂-123 catalysts using *operando* FTIR spectroscopy (Reviewer-Only Figure 1). The IR spectrum of the Pt/TiO₂ exhibits the bands associated with both nitro (1525, 1491, and 1346 cm⁻¹) and vinyl groups (1417, 1447, and 1630 cm⁻¹), indicating simultaneous

adsorption of nitrobenzene and styrene. However, it is important that the IR spectrum of the Pt/Sn-TiO₂-123 includes the bands associated only with the nitro group (1525, 1491, and 1346 cm⁻¹), indicating the selective adsorption of nitrobenzene on the Pt/Sn-TiO₂-123 (the band at 1590 cm⁻¹ is assigned to the aromatic ring), whereby the styrene adsorption is prevented by the nitrobenzene. These results demonstrate a unique feature of the Pt/Sn-TiO₂-123 for selective adsorption of nitrobenzene, which is responsible for a significant enhancement of vinyl aniline selectivity in the hydrogenation of 3-nitrostyrene on the Pt/Sn-TiO₂-123 catalyst.

These results and the corresponding discussion have been added to the revised manuscript as Supplementary Figure 23 and Supplementary Table 6.

Reviewer-Only Table 1. TOFs in the hydrogenation of nitrobenzene and styrene at various ratios of reactant to catalyst (S/C) on the Pt/TiO₂ and Pt/Sn-TiO₂-123 catalysts.

Catalyst	Feed (S/C)		TOF (mol _{converted} mol _{Pt} ⁻¹ h ⁻¹)	
Pt/TiO ₂	2500	0	1206	-
Pt/TiO ₂	0	2500	-	1608
Pt/TiO ₂	1250	1250	1010	905
Pt/Sn-TiO ₂ -123	2500	0	1917	-
Pt/Sn-TiO ₂ -123	0	2500	-	1262
Pt/Sn-TiO ₂ -123	1250	1250	2150	134

Reaction conditions: 10 mmol of substrate, 20 mL of toluene, 318 K, and 0.2 MPa of H₂. The TOFs were calculated on the basis of exposed Pt atoms on the catalysts.

Reviewer-Only Figure 1. *Operando* IR spectra characterizing the competitive adsorption of nitrobenzene and styrene on the Pt/TiO₂ and Pt/Sn-TiO₂-123 catalysts. The catalysts were pre-reduced by treatment in flowing H₂, and then a mixture of styrene and nitrobenzene was introduced. The spectra were recorded after a 10-min period for desorption (of weakly bound species) at 323 K.

Reviewers' Comments:

Reviewer #2 (Remarks to the Author):

The manuscript was revised adequately to my comment.
I would like to recommend its publication.